# The Influence of Corporate Social Responsibility Aspects on Business Model Innovation, Competitive Advantage, and Company Performance: A Study on Small- and Medium-Sized Enterprises in Iran

Mohammadsadegh Omidvar [1] and Maria Palazzo [2,*]

1   Department of Business Administration, Faculty of Management, Kharazmi University, Tehran 1599964511, Iran; m.sadeghomidvar@gmail.com
2   Department of Economics, Mercatorum University, 00186 Rome, Italy
*   Correspondence: maria.palazzo@unimercatorum.it

**Abstract:** This study explores how the dimensions of corporate social responsibility (CSR) as defined by Carroll, along with environmental responsibility, impact business model innovation (BMI), competitive advantage, and firm performance in small- and medium-sized enterprises in Iran. This paper proposes a theoretical framework, based on past studies. Afterward, structural equation modeling was used to test the conceptual model. The data of this research were collected face-to-face, and 483 valid questionnaires were collected from small- and medium-sized businesses in Iran. The results show that all dimensions of CSR (except philanthropic) directly and significantly affect Business Model Innovation. Furthermore, the competitive advantage is significantly impacted by the economic, legal, and ethical aspects of CSR. Additionally, the findings demonstrate that both BMI and competitive advantage play a direct and substantial role in influencing a company's performance. This study represents one of the initial investigations to specifically analyze how each facet of corporate social responsibility influences Business Model Innovation and competitive advantage. It is worth noting that a new dimension, environmental responsibility, was incorporated into Carroll's original model due to the growing significance of environmental concerns. This paper gives managers a better insight into CSR and its effects on company performance. In addition, it shows managers which aspects of CSR can have an impact on BMI and competitive advantage.

**Keywords:** corporate social responsibility; environmental responsibility; business model innovation; competitive advantage; firm performance; SME; Iran

## 1. Introduction

The subject of corporate social responsibility (CSR) has been receiving the attention of university and academic researchers for several years. Over the years, many researchers have investigated whether CSR can affect company performance [1–10]. Thanh et al. [11] investigated the performance of SMEs and showed the direct and significant impact of CSR and firm performance (FP). Bahta et al. [10] investigated this relationship in developing countries (Eritrea) and concluded that CSR significantly affects the performance of SMEs.

In addition, some researchers have concluded that CSR can be a useful and efficient tool for companies to achieve their strategic goals [12–14]. CSR can be a resource for companies that brings non-monetary returns to the company in addition to economic benefits [14].

Today, SMEs play a crucial role in the economy, and it is estimated that about 400 million SMEs produce about USD 36,300 bn per year [15]. For example, in 2019, the number of SMEs in Iran was 43,650 in 800 industrial towns, and 1100 of these companies exported their products to other countries [16]. As a result, the literature needs to

conduct more research on SMEs. In addition, most of the research in this field has been conducted in developed countries, and little research has been performed in emerging economies [17] and developing countries [18]. In the literature, most of the research on CSR has been conducted on large companies, and little of this research has investigated the issue of SMEs. But the same few studies indicate that the presence of SMEs in CSR activities can have many benefits for these companies [19]. Possibly examples of these advantages include more profits [20], more employee loyalty [21], gaining a competitive advantage (CA) [22], improving the business image [23], winning the trust of shareholders and business partners [11], and finally improving SME performance [24].

The literature contains several gaps that need to be addressed. The first gap is that a few studies in this field have considered only commercial Business Model Innovation (BMI). The current body of literature has predominantly centered around the impact of CSR on BMI toward social or sustainable BMs [25]. Social or sustainable business models are mainly aimed at solving social and environmental problems, which are dramatically different from commercial BMs that are mainly oriented toward profit generation [26]. Nevertheless, the majority of businesses prioritize commercial business models and tend to overlook social business models because their primary objective is profit generation. [27]. Hence, the literature falls short of providing a definitive response to the question of whether companies are genuinely inclined to incorporate corporate social responsibility (CSR) into their business models. Another critical void in this field arises from the limited research on the connection between CSR and Business Model Innovation (BMI). Consequently, our understanding of whether CSR yields positive outcomes for BMI remains incomplete, and there has been relatively scant investigation into the impact of CSR on BMI. Consequently, the literature does not offer a conclusive answer as to whether it is indeed imperative for businesses to integrate CSR into their business models [7].

Third, to the best of our knowledge, most research that investigated the relationship between CSR, BMI, and CA considered them as variables, but in this research, we investigate the impact of five dimensions of CSR on BMI, and CA.

To fill these gaps, this study aims to investigate the relationship between CSR activities, CA, and FP in Iranian SMEs. There are different models in the field of CSR, but one of the well-known CSR models is Carroll's pyramid of social responsibility, which has four levels [28–33], and this model has garnered significant interest among researchers in this particular area [29,31,34–42].

In addition, today, the issue of the environment has attracted a lot of attention. Today, people tend to buy from companies whose activities cause the least possible damage to the environment [43]. Also, several researches have investigated the response of consumers toward the acceptance of environmental responsibility by companies and the effect of their adherence to environmental protection [43–46].

There is no doubt that in the competitive business world of various industries, the proper performance of firms, especially the financial performance of firms, is very necessary to survive. Over the past years, many studies have examined the correlation between CSR and FP. In the meantime, many experts have proven the positive correlation between CSR and FP [1–4,47].

Despite the studies conducted in the field of CSR, companies are cautious or even skeptical about CSR investments. The results of this research contribute to the managers who are faced with the dilemma of adhering or not adhering to their social responsibilities and show the results of adhering to these responsibilities. This research shows whether CSR affects BM innovation and improves the financial performance of companies or not.

Thus, this study tries to answer these questions: (i) Is it possible for companies to improve their business models by strengthening the social aspects of their value creation systems or by adopting corporate social responsibility (CSR) strategies to create entirely new business models? (ii) Can active participation in CSR activities lead to the superiority of the firm's competitive position?

The results of this research show company managers whether companies can integrate their resources through CSR, which helps to grow the resource base and their heterogeneity and, as a result, helps in the possible realization of heterogeneous BMI at a lower cost and in providing opportunities for companies to achieve BMI and CA through their CSR and to ultimately improve their performance.

On the other hand, this research is conducted in Iran (a developing country), and according to this issue, the effects of CSR on the mentioned cases in the business environment under the conditions of a developing country are investigated. Finally, it should be stated that this research was carried out in a Muslim country, and this issue can have a different impact on the moral part of CSR and others not covered in research conducted in countries with different conditions. As a result, the results of this research can be of great help to company managers operating in developing and Muslim countries.

In Section 2, we review the existing literature and research and develop hypotheses. Then, in Section 3, the data collected for this study are analyzed, and the hypotheses of this study are examined. In Section 5, we discuss the results obtained from Section 3. The theoretical and managerial implications section explains the theoretical and practical consequences of this research. Finally, we express this research's limitations and suggestions for future research.

## 2. Literature Review

### 2.1. Corporate Social Responsibility (CSR)

In recent decades, the concept of CSR has evolved [48] and different definitions of this concept have been presented over the years [49], but in general, CSR does not have a universal definition [48,50–52]. Refs. [40,53] define CSR as a set of specific practices in that companies prioritize the social good over their personal interests. Companies carry out such activities in order to positively influence their stakeholders, especially customers [54]. In recent years, researchers have investigated the direct and indirect impact of CSR on FP [3–10]. Among the research that has examined the direct and indirect impact of CSR on FP, many of them have concluded that CSR and FP have a correlation [1–4,10,11,47]. Rhou et al. [8] indicate that the relationship between CSR and FP depends on the group's stakeholders' attitude toward social responsibility. This study shows that the positive attitude of the organization's stakeholders toward CSR has a positive effect on FP. Kim et al. [55] state that companies' involvement in CSR activities can significantly increase their market share. Saeidi et al. [56] investigated how CSR contributes to FP. The results of this study, which gathered data from Iranian businesses, demonstrated that the connection between corporate social responsibility (CSR) and financial performance (FP) is entirely mediated. Hu, Zhang, and Yan [7] conducted a study with the aim of investigating the impact of CSR on BMI. The results showed that CSR and organizational legitimacy affect BMI.

Nowadays, the role of CSR on CA has become a special research topic and has gradually attracted more people [57,58]. Researchers have come to the conclusion that CSR can be a tool for companies to gain CA [14,59].

### 2.2. Economic, Legal, Ethical, Philanthropic, and Environmental Responsibility

Companies must be profitable to continue their activities. They need to be profitable in order to buy raw materials, pay workers' salaries, cover other expenses, and pay their operating expenses from their profits [60]. Friedman [61] emphasizes that companies are, first, responsible for themselves,; therefore, they must be profitable for their survival. In addition, a profitable company helps the economic growth of the society in which it operates [53,62,63].

McWilliams and Siegel [5] state that stakeholder collaboration may be beneficial for firms, which is crucial when applying BMI for value production. Hu et al. [7] indicate that CSR has a positive and significant effect on BMI. According to Rochayatun et al. [64], the provision of CSR program aid by corporations or governments to SMEs leads to

a notable enhancement in the competitive edge and long-term viability of these SMEs. Saeidi et al. [56] indicate that economic CSR has direct and significant effects on CA.

Therefore, we propose these hypotheses:

**H1.** *Economic CSR has direct and significant effects onBMI.*

**H2.** *Economic CSR has direct and significant effects onCA.*

Another important facet of CSR is legal responsibility. Legal CSR can be understood as requiring companies to adhere to society's laws [63]. In other words, societies expect the companies that operate within them to abide by their rules [60,65–67].

Ngo and Le [68] claim that because CSR entails creative actions and modifications for generating value, enhancing value, and augmenting current value, it can support BMI. McWilliams and Siegel [5] state that engaging in CSR activities and using its resources can lead to more investment in research and development (R&D), leading to increased innovation. Du et al. [12] argue that CSR is an organizational obligation to contribute to societal and environmental progress, while also attaining a sustainable competitive edge. Kotler and Lee [69] found that CSR helps companies to develop better competitive advantages than other companies. Therefore, we propose the following hypotheses:

**H3.** *Legal CSR has direct and significant effects on BMI.*

**H4.** *Legal CSR has direct and significant effects on CA.*

Having said that, it is important to highlight that every society has a set of ethical norms to which companies are expected to respect and adhere [70]. Carroll states that companies are expected to adhere to these ethical norms and that their activities are not contrary to these ethical norms [30,37,60].

Dunbar et al. [71] claim that CSR positively affects the risk-taking ability of leaders because they understand the value that CSR brings to their business and the first-mover advantage. As a result, CSR can affect BMI because CSR plays an important role in business leaders' risk-taking. McWilliams and Siegel [5] concluded that companies involved in CSR activities gain benefits such as developing products with socially responsible features. Moreover, CSR initiatives have been found to positively contribute to the economic development of nations and play a crucial role in fostering sustainable competitiveness within both the service and industrial sectors [72]. Cegliński and Wiśniewska [73] showed that CSR activities performed by firms can have many benefits for the company, which may turn into a CA. Therefore, we propose the following hypotheses:

**H5.** *Ethical CSR has direct and significant effects on BMI.*

**H6.** *Ethical CSR has direct and significant effects on CA.*

The fourth level of Carroll's CSR pyramid is philanthropic responsibility. Philanthropic CSR involves society's expectations of companies to be good corporate citizens [63]. This aspect of CSR includes activities that go beyond ethical issues and are voluntarily carried out by companies [74] and improve people's quality of life in society [75]. Ngo and Le [68] stated that CSR stimulates positive stakeholder engagement for the business, which is critical to BMI's success. Martinez-Conesa, Soto-Acosta, and Palacios-Manzano [76] investigated the mediating role of innovation between CSR and FP and concluded that innovation performance has a partial mediating effect on the relationship between CSR and FP. The findings of this research also show that CSR is a significant driver for companies to be more innovative. CSR has the potential to facilitate the development of novel markets and attract additional customers, thereby establishing a lasting competitive advantage [77]. Nyuur et al. [57] established that CSR activities strengthen companies' competitiveness. Therefore, we propose the following hypotheses:

**H7.** *Philanthropic CSR has direct and significant effects on BMI.*

**H8.** *Philanthropic CSR has direct and significant effects on CA.*

Finally, environmental CSR is a firm's concern about the environment [78]. In other words, environmental CSR indicates the attempt by a firm to cause the least damage to the environment [43].

According to Halkos and Skouloudis [79], environmental issues put pressure on established business models and create chances for novel business strategies. They showed that company adherence to CSR promotes innovation. In addition, CSR provides firms with opportunities to effectively address their social duties, resulting in the acquisition of durable competitive benefits [80]. Eyasu and Arefayne [81] showed that environment-based CSR has a positive and direct impact on the CA. Therefore, we propose the following hypotheses.

**H9.** *Environmental CSR has direct and significant effects on BMI.*

**H10.** *Environmental CSR has direct and significant effects on CA.*

### 2.3. Business Model Innovation (BMI)

BMI is a term that has emerged recently, and different authors have defined it in different ways. However, the main issue of BMI is related to CA and superior performance [82]. BMI can be defined as a plan that shows how firms can create value and provide this value to customers. In other words, BMI refers to "how the business performs and creates value for shareholders." [83].

BMI has recently attracted the attention of many researchers and academics [77–79]. Anwar [82] investigated the importance of BMI in the performance of SMEs and the mediating role of CA. The findings from this study indicate that the BMI has a clear and notable influence on the performance of both CA and small- and medium-sized enterprises' performance. Also, based on the perspective of resource advantage theory, BMI is an advantaged resource of the enterprise that can result in competitive advantages for the business, so CA mediates the relationship between BMI and SME performance.

Cucculelli and Bettinelli [84] state that updated BMs bring better performance to companies than traditional BMs. Effective BMI can bring a higher level of profit to the company than traditional BMs [85] and bring more return on equity (ROE) for the company [86]. This has caused some companies to change direction from technological innovations to BMI [87]. Research shows that BMI has a direct effect on the performance of SMEs [77,79,88].

**H11.** *BMI positively and significantly affects firm performance.*

### 2.4. Competitive Advantage

Competitive advantage is defined as "a firm's strategic advantages over its competitors in the industry, which enables it to perform against competitors and competing firms" [89]. In today's dynamic market, CA has been given enough attention by companies because it significantly contributes to the company's financial and market performance [90]. Several studies found a relationship between CA and FP [56,77,86].

**H12.** *CA positively and significantly affects firm performance.*

### 2.5. Conceptual Model

The conceptual model of this study is derived from several studies [7,56,57,77,87,89–96] (see Figure 1).

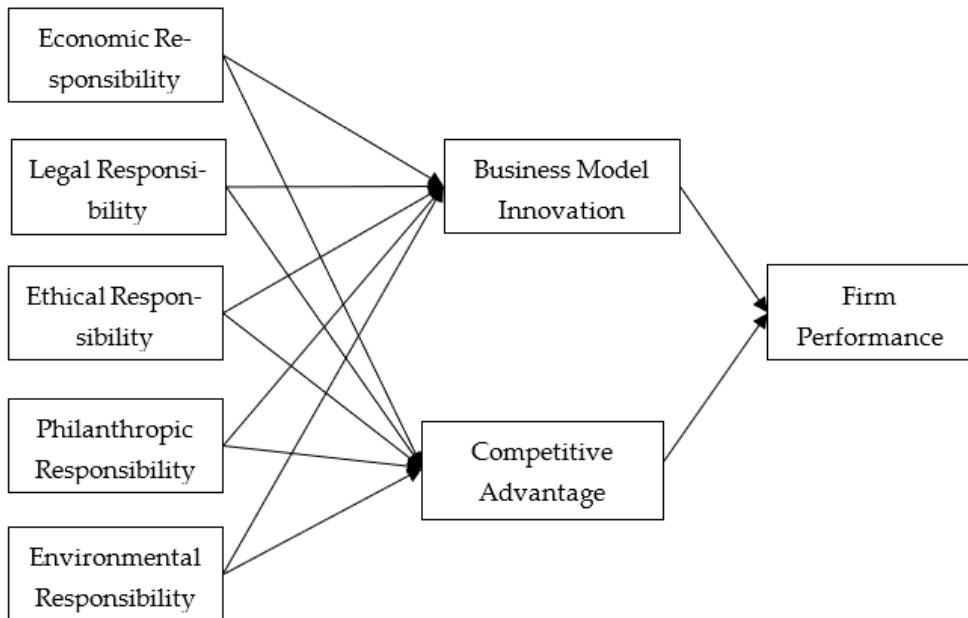

**Figure 1.** A proposed conceptual model. Source: own elaboration.

## 3. Methodology

### 3.1. Research Design, Data, and Sample

This study employed a quantitative research methodology to investigate the associations between CSR dimensions, BMI, CA, and FP. The selection of a quantitative methodology is considered more appropriate for this specific research undertaking for various reasons. First and foremost, this approach facilitates the incorporation of a significant sample size, hence augmenting the generalizability of the results. Additionally, the incorporation of quantitative data enables a more streamlined analysis procedure, enabling a thorough investigation of the factors under consideration. Finally, the utilization of the quantitative approach enables the systematic examination of hypotheses, hence facilitating the development of strong and reliable findings [97].

The samples of this study were collected using convenient sampling. Coordination was conducted with several Iranian SMEs from different industries to collect data of this research face-to-face. A total of 652 questionnaires were distributed among the top employees (owners, senior managers, middle management, and operations management) of these companies, of which 517 (79.29%) were completed. Among these questionnaires, 483 valid questionnaires were extracted.

### 3.2. Measurement

The measurement scales of the constructs were derived from the literature and adapted to the current research context of Iran. It is worth noting that the questionnaire items were originally in English. But due to cultural disparities and the need for transparency, it was then translated into Farsi (Persian). All participants were guaranteed anonymity regarding their opinions and personal identity. Kline [98] suggests that several items should be used to measure a structure instead of one item. Therefore, in this research, three items were used to measure the research structures for each variable, specifically three items for each CSR dimensions based on [53], nine items for BMI based on [7], five items for CA based on [99], and six items for firm performance based on [100].

### 3.3. Data Analysis

According to Galbreath and Shum [101], SEM is better than common regression analysis in the field of CSR. For the statistical analysis, confirmatory factor analysis (CFA) and structural equation model (SEM) were used to test the conceptual model. To analyze

the data, SPSS version 21, AMOS version 24 software, and SmartPLS 4 were used. The SPSS program was only used to enter the data into the AMOS program and to obtain Cronbach's alpha. As AMOS is especially used for structural equation modeling, path analysis, and confirmatory factor analysis, and since the conditions of this research followed the characteristics of the covariance-based test, the AMOS program was used.

The data analysis was developed in two phases. In the first phase, confirmatory factor analysis (CFA) was run to estimate and evaluate the dimensions. Confirmatory factor analysis is a statistical technique used to verify the factor structure of a set of observed variables. Confirmatory factor analysis is often the analytic tool of choice for developing and refining measurement instruments, evaluating the fit of the measurement model and verification of the construct fit indices, assessing construct reliability and validity, and identifying method effects [102]. In the second phase, an SEM analysis was run to test the hypotheses and the model fit.

Most of the CSR studies, especially the study of the impact of CSR on company performance, have been conducted in developing countries [103]. Moreover, Saeidi et al. [56] stated that the issue of CSR is not sufficiently recognized academically and practically in Iran. As a result, there is a need to conduct many studies in this field in Iran. Examining CSR's impact on companies' performance can be useful because CSR has not been sufficiently considered in Iranian businesses and academic environments [104]. In addition, the shareholders of Iranian companies expect a higher level of CSR than what is actually realized by the companies, and this shows that Iranian businesses have not been able to understand or implement CSR expectations well [105].

Therefore, considering that the need for research in the field of CSR and FP is well felt in Iran [103], Iranian manufacturing and service companies have been considered for this research to provide information for both academics and people working in various industries to gain a better view of CSR. Furthermore, considering Iran's status as a developing nation and the limited prior research conducted in the desired field within developing countries, the findings of this study could yield valuable insights.

## 4. Results

Table 1 shows the firms' information. There were 83 owners with 17.2% who participated in this study. From the 483 responses, 145 (30.0%) respondents were senior managers, 145 (30.0%) were in middle management, and 110 respondents (22.8%) were in operations management. A total of 11% of the companies from which data were collected were in the ICT industry, 19.9% of the companies were in the finance and banking industry, 19.9% of the companies were in the pharmaceutical chemical industry, 24.4% of them were in the construction industry, 14.5% of them were in education, and 10.4% of them were in other industries. Seventy-one owners and managers of these companies had 0 to 50 employees, 125 had 51–100 employees, 125 had 101–150 employees, 116 had 151–200 employees, and 46 had 20–250 employees. Three hundred and twenty firms started their operation 10 years ago, 156 firms started operation from 11 to 20 years ago, while 107 firms have been working for more than 21 years.

**Table 1.** Profile of the firms.

| | | Total (N = 483) | |
|---|---|---|---|
| | **N** | | **Percentage (%)** |
| Position | | | |
| Owners | 83 | | 17.2 |
| Senior manager | 145 | | 30.0 |
| Middle management | 145 | | 30.0 |
| Operations management | 110 | | 22.8 |

**Table 1.** *Cont.*

| | | Total (N = 483) | |
|---|---|---|---|
| | **N** | **Percentage (%)** | |
| Industry | | | |
| ICT | 53 | 11.0 | |
| Finance and banking | 96 | 19.9 | |
| Pharmaceutical chemical | 96 | 19.9 | |
| Construction | 118 | 24.4 | |
| Education | 70 | 14.5 | |
| Other | 50 | 10.4 | |
| Size of firms | | | |
| 20–50 employees | 71 | 14.7 | |
| 51–100 employees | 125 | 25.9 | |
| 101–150 employees | 125 | 25.9 | |
| 151–200 employees | 116 | 24.0 | |
| 201–250 employees | 46 | 9.5 | |
| Age of firms | | | |
| 10 years and less | 220 | 45.5 | |
| 11–20 years | 156 | 32.3 | |
| 21 and above years | 107 | 22.2 | |

Source: Research data.

The result of the measurement model test showed a good fit to the data: $\chi^2$ = 729.628, df = 436, $\chi^2/df$ = 1.673, RMSEA = 0.037, PNFI = 0.729, GFI = 0.912, AGFI = 0.893, CFI = 0.922, IFI = 0.924, and TLI = 0.912. Figure 2 depicts the CFA.

Table 2 displays the factor loadings, Cronbach's $\alpha$, composite reliability (CR), and average variance. Before the review, it should be stated that in the data analysis stage, we eliminated two questions considered for BMI (including BMI 1 and BMI 9 questions) and one question considered for CA (including CA 1 question) due to the incompatibility of the answers given to them with other answers and due to inconsistencies in the model.

Based on Table 2, the standardized factor loadings of most items are higher than 0.5, the recommended threshold value by [106]. But the standardized factor loadings of 3 items are between 4 and 5. Based on Guadagnoli and Velicer [107], standardized factor loadings higher than 0.4 are considered stable. Therefore, the standardized factor loadings are all statistically significant. Cronbach's alphas of four items are higher than 0.70, which are higher than the suggested threshold [108]. Cronbach's alphas of four items is between 6 and 7. According to Cronbach [109] and Hajjar [110], a reliable item can be indicated by a Cronbach's alpha score exceeding 0.6. Table 3 also shows the CR and AVE. Based on Hair et al. [106], a CR above 0.7 shows good internal consistency, and as shown in the table, all the numbers related to CR are higher than 0.7. According to Chin [111] and Jr et al. [112], to achieve an acceptable level of convergent validity, the AVE of each latent construct should be higher than or equal to 0.50. Based on the table, the AVE of 3 variables is higher than 0.5, but the AVE of 5 variables is less than 0.5. According to Fornell and Larcker [108], if the AVE is less than 0.5 but the CR is higher than 0.6, the construct's convergent validity is deemed satisfactory. Therefore, the convergent validity of the construct is deemed acceptable.

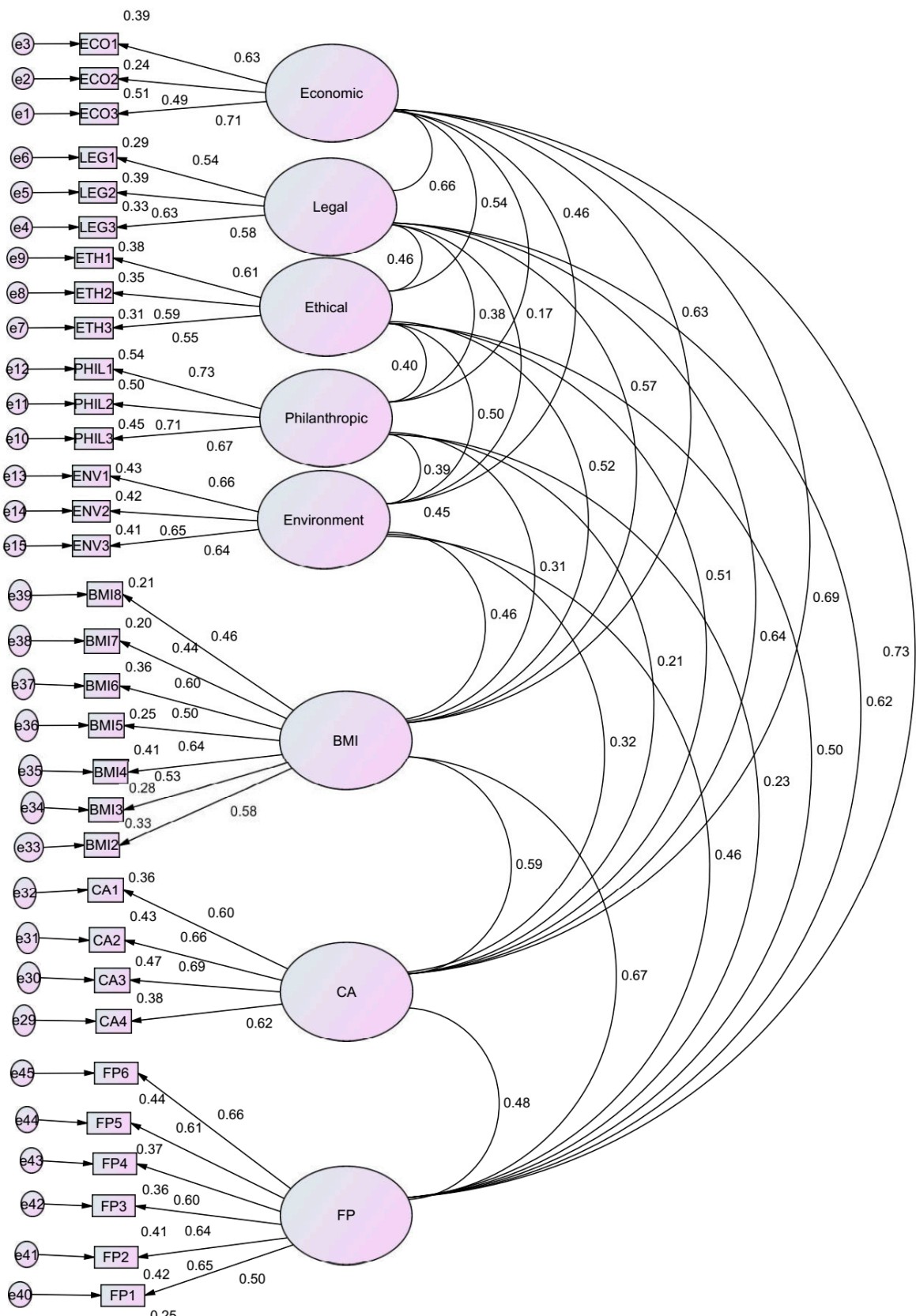

**Figure 2.** CFA result. Source: research data.

**Table 2.** Properties of the measurement model (N = 483).

| Measurement Items | Factor Loading | Cronbach's $\alpha$ | CR | AVE |
|---|---|---|---|---|
| Economic Responsibility | | 0.607 | 0.840 | 0.331 |
| ECO1 | 0.628 | | | |
| ECO2 | 0.493 | | | |
| ECO3 | 0.712 | | | |
| Legal Responsibility | | 0.602 | 0.853 | 0.412 |
| Leg1 | 0.538 | | | |
| Leg2 | 0.628 | | | |
| Leg3 | 0.577 | | | |
| Ethical Responsibility | | 0.612 | 0.880 | 0.530 |
| Eth1 | 0.615 | | | |
| Eth2 | 0.592 | | | |
| Eth3 | 0.555 | | | |
| Philanthropic Responsibility | | 0.747 | 0.933 | 0.597 |
| Phi1 | 0.734 | | | |
| Phi2 | 0.708 | | | |
| Phi3 | 0.674 | | | |
| Environmental Responsibility | | 0.683 | 0.914 | 0.502 |
| Env1 | 0.660 | | | |
| Env2 | 0.651 | | | |
| Env3 | 0.640 | | | |
| BMI | | 0.736 | 0.954 | 0.305 |
| BMI2 | 0.577 | | | |
| BMI3 | 0.530 | | | |
| BMI4 | 0.636 | | | |
| BMI5 | 0.500 | | | |
| BMI6 | 0.597 | | | |
| BMI7 | 0.443 | | | |
| BMI8 | 0.460 | | | |
| CA | | 0.735 | 0.930 | 0.462 |
| CA1 | 0.604 | | | |
| CA2 | 0.658 | | | |
| CA3 | 0.688 | | | |
| CA4 | 0.620 | | | |
| FP | | 0.776 | 0.944 | 0.386 |
| FP1 | 0.497 | | | |
| FP2 | 0.648 | | | |
| FP3 | 0.644 | | | |
| FP4 | 0.604 | | | |
| FP5 | 0.606 | | | |
| FP6 | 0.661 | | | |

Source: Research data.

**Table 3.** Heterotrait–Monotrait Ratio (HTMT).

|  | ECO | LEG | ETH | PHIL | ENV | BMI | CA | FP |
|---|---|---|---|---|---|---|---|---|
| ECO |  |  |  |  |  |  |  |  |
| LEG | 0.661 |  |  |  |  |  |  |  |
| ETH | 0.541 | 0.457 |  |  |  |  |  |  |
| PHIL | 0.173 | 0.385 | 0.405 |  |  |  |  |  |
| ENV | 0.464 | 0.453 | 0.504 | 0.389 |  |  |  |  |
| BMI | 0.696 | 0.641 | 0.507 | 0.211 | 0.316 |  |  |  |
| CA | 0.632 | 0.574 | 0.517 | 0.126 | 0.463 | 0.589 |  |  |
| FP | 0.733 | 0.623 | 0.503 | 0.254 | 0.464 | 0.480 | 0.675 |  |

Source: Research data. Note. ECO = economic CSR; LEG = legal CSR; ETH = ethical CSR; PHIL = philanthropic CSR; ENV = environmental CSR; BMI = Business Model Innovation; CA = competitive advantage; FP= firm performance

In Table 3, the numbers related to the evaluation of [113] are given. Based on [113], an HTMT greater than 0.90 indicates a lack of discriminant validity. As shown in Table 3, all the numbers related to this rate are lower than this amount.

Hypothesis testing and structural equation model.

The estimation results provided a good fit with the data) $\chi^2$ = 779.482; df = 442; $\chi^2$/df = 1.764, RMSEA = 0.040, PNFI = 0.729, GFI = 0.907, AGFI = 0.889, IFI = 0.912, TLI = 0.900, CFI = 0.911). As it is clear from the numbers, the fitting values of these indices are all within an acceptable range based on [106].

Table 4 and Figure 3 show the result of structural equation modeling (SEM).

**Table 4.** Results of the structural equation modeling (n = 483).

| Paths | Standardized Coefficients | t-Value | *p*-Value | Hypotheses |
|---|---|---|---|---|
| H1. Economic CSR has direct and significant effects on BMI. | 0.324 | 3.134 | 0.002 | Supported |
| H2. Economic CSR has direct and significant effects on CA. | 0.415 | 3.730 | 0.000 | Supported |
| H3. Legal CSR has direct and significant effects on BMI. | 0.310 | 3.016 | 0.003 | Supported |
| H4. Legal CSR has direct and significant effects on CA. | 0.336 | 3.332 | 0.000 | Supported |
| H5. Ethical CSR has direct and significant effects on BMI. | 0.226 | 2.571 | 0.010 | Supported |
| H6. Ethical CSR has direct and significant effects on CA. | 0.205 | 2.244 | 0.025 | Supported |
| H7. Philanthropic CSR has direct and significant effects on BMI. | −0.171 | −2.473 | 0.013 | Not Supported |
| H8. Philanthropic CSR has direct and significant effects on CA. | −0.030 | −0.422 | 0.673 | Not Supported |
| H9. Environmental CSR has direct and significant effects on BMI. | 0.162 | 2.222 | 0.026 | Supported |
| H10. Environmental CSR has direct and significant effects on CA. | −0.120 | −1.546 | 0.122 | Not Supported |
| H11. BMI has direct and significant effects on firm performance. | 0.644 | 6.507 | 0.000 | Supported |

**Table 4.** *Cont.*

| Paths | Standardized Coefficients | t-Value | *p*-Value | Hypotheses |
|---|---|---|---|---|
| H12. CA has direct and significant effects on firm performance. | 0.161 | 2.349 | 0.019 | Supported |

Source: Research data. Note. R-square (R2): BMI (0.595); CA (0.592); FP (0.557).

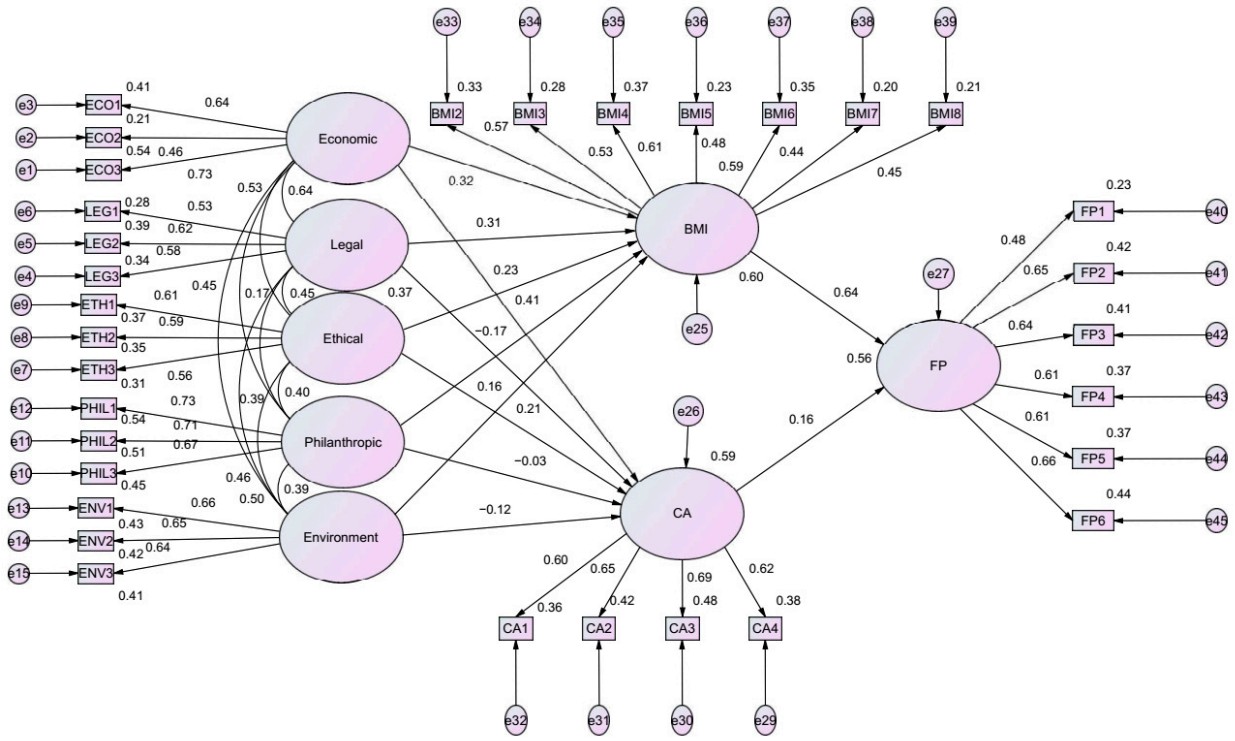

**Figure 3.** Structural equation model result. Source: Research data.

According to the standardized coefficients, t-value, and *p*-value (Table 4 and Figure 3), economic CSR has a direct and significant effect on BMI (β = 0.324, *p* < 0.01) and CA (β = 0.415, *p* < 0.01). These results support hypotheses 1 and 2. Legal CSR has a direct and significant effect on BMI (β = 0.310, *p* < 0.01) and CA (β = 0.336, *p* < 0.01). These results support hypotheses 3 and 4. Ethical CSR has a direct and significant effect on BMI (β = 0.226, *p* < 0.05) and CA (β = 0.205, *p* < 0.05). These results support hypotheses 5 and 6. Philanthropic CSR has a significant inverse effect on BMI (β = −0.171, *p* < 0.05) and does not have a direct and significant effect on CA (β = −0.030, *p* > 0.05). These results reject hypotheses 7 and 8. Environmental CSR has a direct and significant effect on BMI (β = 0.162, *p* < 0.05) but does not have a direct and significant effect on CA (β = −0.120, *p* > 0.05). These results support hypotheses 9 and 10. BMI has a direct and significant effect on FP (β = 0.644, *p* < 0.01) and, as a result, hypothesis 11 is supported. Finally, CA has a significant effect on FP (β = 0.161, *p* < 0.05), and as a result, hypothesis 12 is supported.

Table 4 shows that the R-square (R2) value for BMI was 0.595, which means that the CSR dimension could affect the corporate image variable by 59.5%, while the remaining 40.5% was the influence of other variables, which was not included in this study. The R-square (R2) value for CA was 0.592, which means that the CSR dimension could affect the service quality variable by 59.2%, while the remaining 40.8% was the influence of other variables, which was not included in this study. Finally, the R-square (R2) value for firm performance was 0.557, which means that the CSR dimension could affect the customer retention variable by 55.7%, while the remaining 44.3% was the influence of other variables, which was not included in this study.

## 5. Discussion

The aim of this study is to show the impact of different dimensions of CSR on BMI and CA and ultimately company performance. The framework and results of this company can help the literature as well as managers of different companies to make decisions about CSR strategies. The results of this research help managers, especially managers of small and medium Iranian companies, to gain a better understanding of CSR. This research shows which dimensions of CSR have a greater impact on BMI and the company's CA.

The result is that economic CSR directly and significantly affects BMI and CA. It is clear that the more companies that focus on their profitability, the more they will pay attention to their BM and the BMI. These results are consistent with Hu et al. [7], showing that CSR has a direct and signification effect on BMI. It can also be stated that greater profitability has a two-way relationship with CA, as the greater the profitability of a company, the greater the company's advantage over its competitors, and the greater the company's CA over its competitors. In the market, it will be more profitable. As a result, it can be stated that whenever a company focuses on profitability, it will seek to gain CA, and as a result, it is more likely to gain CA over its competitors. This is consistent with Saeidi et al. [56] that states that economic CSR positively and significantly affects CA. This is also consistent with Rochayatun et al. [64], who stated that CSR programs lead to a significant increase in competitive advantage.

The results indicate that legal CSR directly and significantly affects BMI and CA. This can be because companies that adhere to the laws of a society can be supported by the people, the government, and even the employees of that company, which can have a significant impact on BMI and CA. This is consistent with McWilliams and Siegel [5] that states that paying attention to CSR issues causes companies to be more involved in research and development (R&D). This is also consistent with Ngo and Le [68] that states that CSR programs can support BMI. On the other hand, the results of this research are consistent with Kotler and Lee [69] that states that CSR helps companies to develop a better CA than other companies. This is also consistent with Du et al. [12] that argues that CSR is an organizational commitment that helps companies achieve a sustainable competitive advantage.

The results show that ethical CSR directly and significantly affects BMI and CA. These results are consistent with Dunbar et al. [71], which claims that CSR leads to an increase in BMI by increasing the risk-taking ability of leaders. These results are also consistent with Cegliński and Wiśniewska [73] that show that CSR activities carried out by companies can have many benefits for the company, which may turn into a CA.

The results show that philanthropic CSR does not have positive and significant effects on BMI and CA. The reason for this could be that philanthropic responsibilities are still not well understood in Iran and people do not have a correct view of this aspect of CSR. Also, companies sometimes make mistakes in recognizing the difference between ethical CSR and philanthropic CSR. On the other hand, due to Iran's unfavorable economic conditions, companies do not have much ability to spend on philanthropic CSR, and this is one of the cases where there is not enough understanding of this aspect of CSR and the results of adhering to it. These results are contrary to Ngo and Le [68] that states that CSR stimulation is critical to BMI success. This result is also contrary to Martinez-Conesa et al. [76] that indicates that CSR is a significant driver for companies to be more innovative. These results are also contrary to Nyuur et al. [57] and Khanzad and Gooyabadi [77], which show that CSR activities strengthen companies' competitiveness.

The results show that environmental CSR directly and significantly affects BMI, but does not have positive and significant effects on CA. This can be because companies that pay attention to the environment are looking for ways to make their operations less harmful to the environment, and as a result, they put more effort into being innovative and finally to have BMI. This is consistent with Halkos and Skouloudi [79], which states that company adherence to CSR promotes innovation. But the results of this research are in conflict with

Eyasu and Arefayne [81], which states that environment-based CSR has a direct impact on CA.

The results also show that BMI directly and significantly affects FP. This result is consistent with research that shows that BMI has a direct effect on SME performance [77,79,88]. This study also shows that CA directly and significantly affects FP. This result is consistent with studies that found a significant relationship between CA and FP [56,77,86].

## 6. Conclusions

Attention to CSR is increasing all over the world [114], because company managers have realized that CSR can bring many benefits to companies [19]. For example, it should be stated that during the past years, many studies have investigated the relationship between the involvement in CSR activities and innovation, and the number of these types of studies is increasing. Also, company managers have realized that the relationship between companies and shareholders has become an essential issue for the success of companies among competitors [115]. As a result, this research was conducted with the aim of investigating the relationship between Iranian SMEs' involvement in CSR activities, BMI and CA, and ultimately the company's performance. The results of this research show that all dimensions of CSR (except philanthropic) affect BMI. Therefore, a significant effect of CSR on BMI can be seen in Iranian SMEs. Also, the results of this research show that most aspects of CSR (except philanthropic and environmental) affect CA. Therefore, it can be said that the involvement of Iran's SMEs in CSR bankruptcy has a significant impact on their CA. As a result, according to the specific results of this research, SMEs in emerging economies like Iran can be recommended to pay special attention to CSR and their shareholders.

### 6.1. Theoretical and Managerial Implications

Current research supports the relationship of responsibilities (economic, legal, ethical, and environmental) with BMI. In addition, the relationship of responsibilities (economic, legal, and ethical) with a CA is supported. Moreover, this study supports the relationship between BMI, CA, and FP. Also, this research's findings support a relationship between CSR dimensions and FP (mediated by BMI and CA) and provide more insight into the literature on the impact of CSR on FP. Furthermore, in this study, the impact of each dimension of CSR on the BMI and CA has been investigated, while in most previous studies that have been conducted to investigate the impact of CSR on these issues, CSR has been discussed as a whole. Therefore, the outcome of this study clearly demonstrates which dimension of CSR can affect BMI, CA, and FP. Second, despite much research on CSR, little research has empirically examined the impact of CSR on BMI, CA, and FP in relation to SMEs, especially in developing countries. Thirdly, in most previous research that investigated the relationship between different aspects of CSR and BMI, CA, and FP, CSR dimensions were included (economic, legal, ethical, and philanthropic), and the environmental dimension was considered as a part of ethical CSR and was discussed. In this research, environmental CSR is considered the fifth dimension of CSR. In summary, this research sets itself apart from other studies on corporate social responsibility by examining the link between CSR dimensions, BMI, CA, and financial performance through a survey. Fifth, little research has been conducted regarding the impact of CSR on small- and medium-sized enterprises (SMEs), making this an area ripe for exploration for Iranian researchers and countries with similar circumstances to Iran. In Iran, much like numerous other Middle Eastern nations, SMEs constitute the majority in terms of numbers and hold significant potential for contributing to the economy. Nevertheless, due to their SME status, there is a scarcity of statistical data and research in these geographical regions. Consequently, conducting research on this distinct business category is imperative for the advancement of CSR. Considering these initial observations, it is evident that our findings must be approached with caution and applying them to larger corporations necessitates thoughtful consideration. Consequently, there is room for future research focusing on large enterprises, particularly in developing nations.

In addition, the findings of this study provide noteworthy information to Iranian managers. First, it was shown in this study that there is a direct and significant relationship between economic CSR, BMI, and CA. This shows that companies should shift a significant part of their focus onto profitability and finding new and different ways to earn profit from their competitors in the market. Also, the results of this research show that legal CSR has a significant positive effect on BMI and CA. As a result, company managers should pay special attention to complying with the laws set by the communities in which they operate. These results show that company managers should know the laws of the countries in which they operate and follow them. It should be noted that non-compliance with the laws of the countries may lead to dissatisfaction of the people and especially the governments toward the company, which can have bad consequences for the company. In addition, this research shows the effect of ethical CSR on BMI and CA. Company managers must respect the moral norms specified by any society. Paying attention to these norms can make the people and associations of each society support that company more. This research also shows that environmental CSR has a positive and significant effect on BMI. This shows that company managers should pay attention to environmental issues. They should plan in such a way that during the process of creating the value of their company, they cause the least damage to the environment.

*6.2. Limitations and Future Research Direction*

This research is not without limitations. First, the findings of this study are limited to Iran. The research review can be conducted in several countries to achieve more general results. Iran is a developing economy; future research can be performed in developed economies and their results can be compared.

Secondly, in this research, we investigated the impact of CSR dimensions on BMI and CA. Therefore, future research can examine these dimensions in other aspects of firm performance.

Fourth, the data of this study were collected using a questionnaire. Although this method has been used by many other researchers in this field, the use of questionnaires may lead to measurement bias. As a result, future research could address potential biases from other data collection tools and thereby address this limitation.

**Author Contributions:** Conceptualization, M.O. and M.P.; methodology, M.O.; software, M.O.; validation, M.O.; formal analysis, M.P.; investigation, M.O.; resources, M.P.; data curation, M.O.; writing—original draft preparation, M.O. and M.P.; writing—review and editing, M.P.; visualization, M.O.; supervision, M.P.; project administration, M.O.; funding ac-quisition, M.P. All authors have read and agreed to the published version of the manuscript.

**Funding:** This research received no external funding.

**Institutional Review Board Statement:** Not applicable.

**Informed Consent Statement:** Informed consent was obtained from all subjects involved in the study. Approval for the study was not required in accordance with local/national legislation.

**Data Availability Statement:** Data available on request.

**Conflicts of Interest:** The authors declare no conflict of interest.

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
