# Peer review of "The Influence of Corporate Social Responsibility Aspects on Business Model Innovation, Competitive Advantage, and Company Performance: A Study on Small- and Medium-Sized Enterprises in Iran"

_sustainability, doi:10.3390/su152215867_

Round 1

Reviewer 1 Report

Comments and Suggestions for Authors

It an interesting piece of work focusing on the impact of CSR on BMI, CA, and FP. It help to understanding the different impact of various dimensions of CSR on BMI and CA. There are seveal points require further explanation. 

1. The development of hypotheses require strong theoretical and expirical discussion. 

2. The sampling process needs further introduction to improve representativeness.

3. The AVE values are quite low for some constructs.

4. The manuscript requires proofreading to avoid spelling errors and format problems. 

Considering forementioned issues, the current manuscript reqiures majoe revision.

Comments on the Quality of English Language

The manuscript requires proofreading to avoid spelling errors and format problems. 

Author Response

Dear Reviewers,

We would like to kindly thank you for your evaluation and for the constructive and copious suggestions which have helped us to improve the draft significantly. All your comments and suggestions have been taken into account in the revised paper, as described in the following table.

Any revisions to the manuscript is highlighted in the text, such that any changes is easily reviewed by editors and reviewers.

Best Regards

the authors

Reviewer 2 Report

Comments and Suggestions for Authors

I have a few recommendations.

1. The abstract should have a clearer structure: the main topic/objective of the research, the methodology used, the results and the novelty/usefulness of the research. You have all now except the method but the order is also important. 

2. [12]–[14] should be [12-14]. Check the guidelines and your paper for correcting these. Also these [29], [31], [34]–[42] should be [29,31,34–42].

3. Rhou et al. (2016) should have their number in brackets, not the year. Rhou et al. [number of the reference]. Check the entire paper.

4. In the literature review you have very short paragraphs. Try to group them in bigger paragraphs if they treat the same problem.

5. Explain the abbreviation for BMI the first time it appears in the text. Only afterwards, you can use the abbreviation. So do for this as you did for the others. You do it only in 2.7 but you should do that the first time you use the term and you used it before 2.7

6. The method and the conceptual model you used, also the results are well explained. For the method, did you use only SPSS version 21 242 and AMOS version 24 software? Some parts of the results seem to be obtained with SmartPLS or similar. 

6. In the Discussion, you should show the results of other authors for each of your hypotheses. If they reached or not the same findings. 

7. I advise moving the theoretical and managerial implications to the Conclusion section. Keep the title Conclusion and include paragraphs for the implications, limitations and future research directions.

8. Even if you mention the use of SMEs as a limit, I do not know the situation in your country, but in mine, SMEs are the majority as a number, being an important part of the economy. Also, because they are SMEs, the statistics and research in my country are not so high so having research on SMEs is a big plus for me. If the situation is similar in your country and there is not so much attention given to SMEs, I would put it in the novelty part of your research, not limitation. 

9. The references should be formatted using the template provided on the journal website. They are not according to it now. 

Congrats for your work and great success!

Author Response

Dear Reviewers,

We would like to kindly thank you for your evaluation and for the constructive and copious suggestions which have helped us to improve the draft significantly. All your comments and suggestions have been taken into account in the revised paper, as described in the following table.

Any revisions to the manuscript is highlighted in the text, such that any changes is easily reviewed by editors and reviewers.

Best Regards

the auhtors

Round 2

Reviewer 1 Report

Comments and Suggestions for Authors

Thanks for responding to my concerns and make relavant changes in the revised manuscript. I have no further concers.

One suggestion. Please check the following citation: 

Tiep Le, T.Ngo, H.Q. and Aureliano-Silva, L. (2023), "Contribution of corporate social responsibility on SMEs' performance in an emerging market – the mediating roles of brand trust and brand loyalty", International Journal of Emerging Markets, Vol. 18 No. 8, pp. 1868-1891. https://doi.org/10.1108/IJOEM-12-2020-1516

Author Response

Dear Reviewers,

We would like to kindly thank you for your evaluation and for the constructive and copious suggestions which have helped us to improve the draft significantly. All your comments and suggestions have been taken into account in the revised paper.

Best Regards

the authors

Reviewer 2 Report

Comments and Suggestions for Authors

Dear authors,

Congratulations on your work and the improvement done to the paper. 

I am glad that I was able to help a little bit with making your paper better.

I wish you great success with your research career. 

Author Response

Dear Reviewer,

We would like to kindly thank you for your evaluation and for the constructive and copious suggestions which have helped us to improve the draft significantly. 

Best Regards

the authors
